# Equidistant Landmarks Fail to Produce the Blocking Effect in Spatial Learning Using a Virtual Water Maze Task with Healthy Adults: A Role for Cognitive Mapping?

**DOI:** 10.3390/brainsci15040414

**Published:** 2025-04-19

**Authors:** Róisín Deery, Seán Commins

**Affiliations:** Psychology Department, Maynooth University, W23 F2K8 Kildare, Ireland; roisin.deery@mu.ie

**Keywords:** blocking, spatial learning, human, virtual water maze, cognitive map, associative learning

## Abstract

Background/Objectives: Cue competition is a feature of associative learning, whereby during learning, cues compete with each other, based on their relative salience, to influence subsequent performance. Blocking is a feature of cue competition where prior knowledge of a cue (X) will interfere with the subsequent learning of a second cue (XY). When tested with the second cue (Y) alone, participants show an impairment in responding. While blocking has been observed across many domains, including spatial learning, previous research has raised questions regarding replication and the conditions necessary for it to occur. Furthermore, two prominent spatial learning theories predict contrary results for blocking. Associative learning accounts predict that the addition of a cue will lead to a blocking effect and impaired performance upon testing. Whereas the cognitive map theory suggests that the novel cue will be integrated into a map with no subsequent impairment in performance. Methods: Using a virtual water maze task, we investigated the blocking effect in human participants. Results: Results indicated that the cue learned in phase 1 of the experiment did not interfere with learning of a subsequent cue introduced in phase 2. Conclusions: This suggests that blocking did not occur and supports a cognitive mapping approach in human spatial learning. However, the relative location of the cues relative to the goal and how this might determine the learning strategy used by participants was discussed.

## 1. Introduction

Cognitive map theory [1] provides an influential account of spatial navigation and memory, whereby animals can develop a map-like representation of their environment through exploration. Associations between different landmarks and between landmarks and the goal (allocentric relationships), as well as between the navigator, landmarks and/or the goal (egocentric relationships) can be integrated into a cognitive map. An alternative account, based on associative learning theories (e.g., see [2]), may also provide an equally valid account of spatial learning. An important feature of associative learning accounts is that an event may be preceded by significant cues that are used by the learner to predict an outcome [3]. This can be applied to spatial learning whereby some landmarks/cues, due to their characteristics, e.g., size, shape, color, brightness, and especially proximity, are more salient or noticeable than other cues. These cues compete with each other to gain influence over behavior [4,5]. Cue competition gives rise to associated effects such as overshadowing (see [6]) and blocking. 

Blocking, as originally described by Kamin [7,8,9], was observed when rodents were trained to respond to element A and subsequently trained to respond to the same element A with the addition of element B. When tested with element B on its own, rodents failed to respond to it, leading to the conclusion that the prior learning of element A on its own had interfered with or had blocked the learning of element B. Importantly, cognitive map theory and associative learning theory both predict different outcomes with respect to blocking; as such, the phenomenon has been used to test both theories (see [10,11]). For example, the cognitive map theory suggests that if an environment has been learned and new spatial information is then provided (e.g., an additional landmark), the internal representation of the environment will be updated automatically through attention and exploration. Specifically, if subjects are initially trained in an environment with a number of landmarks (that can be used to indicate a particular location), then if another landmark is added to the original environment, subjects readily update their cognitive map and can use the new landmark to find the goal, provided that the additional landmark has been noticed. Therefore, cognitive map theory predicts that blocking will not be observed. In contrast, associative learning theories predict that the original landmarks will interfere and block further learning of any additional landmarks.

The blocking effect has been found in many non-human animal studies across different learning paradigms [12,13,14,15,16,17], including spatial learning [18,19]. However, studies investigating blocking in human participants in recent years have been limited and mainly focused on the role of geometric cues provided by the environment. For example, such studies have examined whether geometric cues (e.g., the overall shape of the environment) can block non-geometric cues (e.g., landmarks within the environment) [20,21,22,23,24], and these studies have led to mixed results [25], with some showing a blocking effect and others not [26,27,28,29,30]. Although there have been relatively few studies that have focused on exploring the blocking effect using landmarks alone, the results of these have also been mixed. For example, using a virtual water maze task, Jacobs et al. [31] showed that the removal of a subset of distal cues did not impair performance, suggesting that a cognitive map was formed with all landmarks. In contrast, Hamilton and Sutherland [32] showed that blocking can occur. The authors showed that participants were unable to find the hidden target using just the additional cues. This suggests that blocking had occurred, and learning with the original cues interfered with learning about any additional cues. Hardt et al. [10] attempted to explain the difference between these two contrasting findings in a number of experiments. In one experiment, the authors showed that the location of the cues during learning was important. Deletion of a subset of cues close to the target impaired performance, whereas performance was unaffected with removed cues that were further away, suggesting that only a subset of the distal cues was explored and encoded (also see [6]). The authors attributed this to the participants forming an egocentric representation rather than seeing it as a challenge to the cognitive map theory. In a second experiment, the authors highlighted that the provision of instructions may have moderated the blocking effect. Combining the results, the authors suggest that blocking in human spatial learning may be a result of poor performance rather than a learning deficit per se [10].

Given the relative lack of research examining blocking in human spatial learning, particularly with respect to individual landmarks, and the inconclusive nature of the results to date, we set out to further explore the blocking phenomenon using a virtual water maze task (VWM) [33]. We have previously found an overshadowing effect in spatial learning [6]; if we also showed a blocking effect, such a finding would further lend support to the associative learning theory of spatial learning rather than the cognitive map theory. Using a paradigm similar to Hamilton and Sutherland [32], we hypothesized that we should observe a blocking effect. Specifically, participants are required to learn the location of a target using a single cue in phase 1, then a second cue is added in phase 2; we hypothesize that when this group is subsequently tested with only the additional cue, the performance of this group (Blocking group) will be impaired compared to two control groups.

## 2. Materials and Methods

### 2.1. Participants

Participants (n = 60) aged 18–41 (Mean = 21.28, SD = 3.34) were recruited using convenience and snowball sampling and consisted primarily of Maynooth University students. These included 23 males and 37 females. Participants were provided with an information sheet outlining the experiment. All participants gave informed consent prior to starting the experiment and were fully debriefed afterwards. Some of the participants from Maynooth University received course credit for participation. Every participant had a normal or corrected-to-normal vision. Participants reporting severe visual impairments, a history of psychological/neurological impairment, a history of motion or simulator sickness, epilepsy or memory issues, reported a history of drug or alcohol abuse, or were taking psychoactive medication were excluded from the study. A priori power calculations were carried out to estimate the number of participants required to determine the main effect of blocking. Using fixed effects ANOVAs and an effect size of 0.3 with a power of 0.9, *p* = 0.05 and three groups, we estimated 60 participants.

To match participants on visual attention, visual-spatial, and executive functioning, trail-making test [34] part A and B were administered. Part A required the participant to connect numbered circles in ascending order as quickly as possible. In part B, a letter was introduced (1-A, 2-B, 3-C, etc). Participants were required to connect the circles containing these numbers and letters in ascending order whilst being timed by the experimenter. Lower time scores reflected better performance on the task.

### 2.2. Spatial Navigation Task

NavWell [33], a virtual version of the Morris water maze task [35], was used in this experiment. This task required participants to navigate around a virtual arena to locate a hidden target. This target only became visible if the participant traversed it. Participants were then required to recall the target’s location in later trials. The hidden target was positioned in the center of the northeast quadrant of the circular pool for all trials. To aid recall, different configurations of cues were positioned on the arena wall depending on the group and phase of learning (see Figure 1a and below for details).

### 2.3. Procedure

Participants took part in two learning phases. They were randomly assigned to one of three groups (n = 20/group): a Blocking group, a Control 1 and a Control 2 group. Participants in the Blocking group undertook phase 1, which consisted of 8 learning trials (60 s each) to find a hidden target using a bright light cue positioned on the north perimeter wall of the arena (Figure 1b). Phase 2 of their learning trials consisted of a further 8 trials (60 s each) of training to find the hidden target, this time with the previous bright light cue and a second additional cue, a green square, positioned on the east perimeter wall. Both cues were equidistant from the hidden target. If participants did not find the target on a particular trial, they were transported to the target location. All participants were allowed 10 s at the target location before commencing the next trial. The time taken to reach the target and path length scores on each trial were recorded.

Participants in Control 1 group did not participate in phase 1 but simply undertook 8 phase 2 learning trials, where the two cues were available (bright light in the north and green square in the east) (see Figure 1c). Participants in Control group 2 undertook 8 learning trials in phase 1 where they trained to find a hidden target with a novel cue (small green triangle) positioned on the north perimeter wall of the arena. Participants immediately undertook a further 8 trials in phase 2 with the two cues (as above—see Figure 1d). Following the learning phase, all participants from all groups completed the TMT and a single 60 s test trial. For the test trial, all participants were given a single trial of 60 s to recall the location of the target. For this trial, a single cue (green square on the eastern wall) was used (see Figure 1e). The platform did not become visible to the participant during this trial. Following this, all participants completed a brief questionnaire gathering demographic information and details of their experience of using the NavWell task.

### 2.4. Design

This study involved 2 learning phases (learning phase 1 and learning phase 2), where an additional cue was added in learning phase 2. Data from each learning phase were analyzed separately. Within each phase, a mixed factorial design was used where Trial in (8 trials) was the within-group factor and Group (2 groups in learning phase 1: Blocking and Control Group 2; 3 groups in learning phase 2: Blocking, Control group 1, and Control group 2) was the between-group factor. Time and distance to the target in each trial were both used as dependent variables for each of the learning phases. Hence, the design was a 2 × 8 mixed factorial in learning phase 1 and a 3 × 8 mixed factorial in learning phase 2.

In the test phase, quadrant (4 levels; NE, SE, SW, NW) was the within-group factor, and Group (3; Blocking, Control group 1 and Control group 2) was again used as the between-group factor. The mean percentage time (of 60 s) spent in each quadrant was used as the dependent variable. Hence, for the test phase, the design was a 4 × 3 mixed factorial.

### 2.5. Ethical Considerations and Data Analysis

All experiments were approved by the Maynooth University Ethics Committee (BSRESC-2021-2453422, approved Nov. 2022) and were conducted according to the ethical guidelines provided by the Psychological Society of Ireland (PSI). All participants provided their consent. The time to target, path length and percentage of time spent in the target quadrant were extracted from NavWell and imported into Microsoft Excel. Means and standard error of means (SEMs) were calculated for each trial and for each group. Graphs were created using Microsoft Excel. Scores were then imported into IBM SPSS version 26 for analysis. One-way ANOVAs were used to analyze age and TMT scores across groups. Mixed factorial ANOVAs were used to analyze learning and test phases. Where relevant, the Tukey HSD test was used for between-group post-hoc comparisons, and Bonferroni corrected *t*-tests were used for within-group comparisons. A star-based level of significance was used where * *p* < 0.05, ** *p* < 0.01 and *** *p* < 0.001.

## 3. Results

### 3.1. Demographics

We initially compared the three groups to ensure that they were generally matched in terms of age and cognitive abilities. A one way between groups ANOVA was conducted to determine if age differed between groups. The assumption of homogeneity of variances was not violated. There was no statistically significant effect found (F (2, 57) = 3.87, *p* = 0.026). Additionally, a one way between groups ANOVA was conducted to determine if TMT b-a scores differed between groups. The assumption of homogeneity of variances was not violated. There was no statistically significant effect found (F (2, 57) = 0.437, *p* = 0.648). This suggested that groups were generally matched on both age and cognitive abilities (see Table 1).

### 3.2. Learning Phase: Latency


*Phase 1*


During the first phase, the Blocking group (those with the bright light cue (located in the north position) and the Control 2 group (those with the small green triangle cue located in the north position) were required to find the hidden target across 8 trials. Both groups learned the task well, with the Blocking group reducing their time to target from 39.60 s, SD = 21.59 on trial 1 to 14.90 s, SD = 10.94 on trial 8. Similarly, the Control 2 group reduced their time from 48.10 s, SD = 17.12 on trial 1 to 14.40 s, SD = 4.45 on trial 8. A 2 × 8 mixed factorial ANOVA was conducted to analyze the time taken to reach the target for the two groups during Phase 1 learning trials (1–8). There was a significant main effect for trial (F (7, 266) = 21.32, *p* < 0.001, ƞp^2^ = 0.359). Bonferroni corrected *t*-tests revealed that participants were quicker to find the target on T8 compared to T1. There was no statistically significant effect for the group (F (1, 38) = 0.162, *p* = 0.690, ƞp^2^ = 0.004) or trial X group interaction effect (F (7, 266) = 1.896, *p* = 0.070, ƞp^2^ = 0.048).


*Phase 2*


During the second phase, all three groups: Blocking, Control 1, and Control 2, had the bright light cue in the north position and the green square cue in the east position and were again required to find the hidden target across 8 trials. A 3 × 8 mixed factorial ANOVA was conducted to analyse the time taken to reach the target between groups 1 2, and 3 during Phase 2 of participant’s learning trials (9–16). There was a significant main effect for trial (F (7, 399) = 45.178, *p* < 0.001, ƞp^2^ = 0.442). Bonferroni corrected t-tests revealed that participants were quicker to find the target on T16 compared to T9. There was a significant main effect for group (F (2, 57) = 10.366, *p* < 0.001, ƞp^2^ = 0.267) with post hoc Tukey tests revealing that the Blocking group performed significantly better than Control group 1 (*p* = 0.001) and Control group 1 performed significantly better than Control group 2 (*p* = 0.013). There was a significant trial X group interaction effect noted (F (14, 399) = 4.462, *p* < 0.001, ƞp^2^ = 0.135. A further one-way ANOVA was conducted to analyze the differences in time to target between the three groups on trials 9 and 10. There was a statistically significant effect found for trial 9 (F (2, 57) = 9.74, *p* < 0.001) with a post hoc Tukey test revealing that the Blocking group took significantly less time to reach the target (mean = 21.10, SD = 17.13) compared to Control group 1 (mean = 47.30, SD = 17.54) and Control group 2 (mean = 33.30, SD = 21.37). There was a statistically significant effect found for trial 10 (F (2, 57) = 4.74, *p* = 0.012) with a post hoc Tukey test revealing that Control group 1 (mean = 22.50, SD = 17.04) took significantly more time at reaching the target compared to the Blocking group (mean = 12.20, SD = 12.79) and Control group 2 (mean = 10.60, SD = 8.53 (see Figure 2).

### 3.3. Learning Phase: Distance Traveled


*Phase 1*


As a second measure of learning, we examined the distance it took for participants to reach the target. During the first phase, both the Blocking and Control group 2 learned the task and reached the target with a reduced distance. The Blocking group reduced their distance from 171.97 vm (virtual meters), SD = 103.35 on trial 1 to 63.59 vm, SD = 35.43 on trial 8. Similarly, the Control group 2 reduced the distance from 196.21vm, SD = 86.24 on Trial 1 to 67.52 vm, SD = 25.15 on trial 8. A 2 × 8 mixed factorial ANOVA was conducted to analyze path length between the Blocking group and Control group 2 during Phase 1 (1–8). There was a significant main effect for trial (F (7, 266) = 19.571, *p* < 0.001, ƞp^2^ = 0.340 with Bonferroni corrected *t*-tests revealing that all participants showed a significant reduction in path length between trial 1 (mean = 184.09 vm, SD = 94.76) and trial 8 (mean = 65.56 vm, SD = 30.39). There was no significant trial X group interaction effect (F (7, 266) = 1.800, *p* = 0.087, ƞp^2^ = 0.045) or significant group effect (F (1, 38) = 0.716, *p* = 0.403, ƞp^2^ = 0.018).


*Phase 2*


As above, a 3 × 8 mixed factorial ANOVA was conducted to analyze the distance traveled to the target between the Blocking, Control 1, and Control 2 groups during Phase 2 of participant’s learning (trials 9–16). There was a significant main effect for trial (F (7, 399) = 54.154, *p* < 0.001, ƞp^2^ = 0.487) with Bonferroni corrected *t*-tests revealing that all participants showed a significant reduction in path length between trials 9 (mean = 152.24 vm, SD = 99.78) and trial 16 (mean = 66.49 vm, SD = 23.21). There was a significant main effect for group (F (2, 57) = 7.567, *p* = 0.001, ƞp^2^ = 0.210). There was a significant trial × group interaction effect noted (F (14, 399) = 5.373, *p* < 0.001, ƞp^2^ = 0.159). A further one-way ANOVA was conducted to analyze the differences in distance traveled to the target between the three groups on trials 9 and 10. There was a statistically significant effect found for trial 9 (F (2, 57) = 8.59, *p* = 0.001). Post hoc Tukey tests revealed that the Blocking group’s path length scores were significantly shorter (mean = 95.73 vm, SD = 81.77) than Control group 1 (mean = 212.22 vm, SD = 84.35). There was also a statistically significant effect found for trial 10 (F (2, 57) = 6.15, *p* = 0.004). Post hoc Tukey tests revealed that the Blocking group’s path length scores were significantly shorter (mean = 50.70 vm, SD = 34.92) than Control group 1 (mean =103.53 vm, SD = 72.00), see Figure 3).

### 3.4. Test Phase

Following the learning phase, all participants were given a single test trial where only one cue was available to them (green square). A 4 × 3 mixed factorial ANOVA was conducted to analyze the mean percentage time (of 60 s) spent in each quadrant across groups during the retention trial. There was a significant main effect for quadrant (F (3, 171) = 65.274, *p* < 0.001, ƞp^2^ = 0.534) with participants spending significantly more time in the NE target quadrant (mean = 53.15%, SD = 22.45) compared to the NW quadrant (mean = 16.60%, SD = 14.43, *p* = 0.854), SE quadrant (mean = 11.32%, SD = 13.20, *p* = 0.339), and SW quadrant (mean = 18.80%, SD = 11.07, *p* = 0.171). There was no significant quadrant X group interaction effect noted (F (6, 171) = 1.258, *p* = 0.280, ƞp^2^ = 0.042). Importantly, there was no significant main effect for group (F (2, 57) = 0.748, *p* = 0.478, ƞp^2^ = 0.026). All three groups spent a similar amount of time searching for the target in the NE (target) quadrant: Blocking group (54.35%, SD 5.15), Control group 1 (58.85%, SD 4.38), and Control group 2 (46.25%, SD 5.31). See Figure 4.

## 4. Discussion

Contrary to what we hypothesized we did not observe a blocking effect using a virtual water maze task with human participants. All groups searched equally in the target quadrant during the test phase. Our findings are contrary to those of Hamilton and Sutherland [32] and suggest that the cue learned during phase 1 did not interfere with learning the cue introduced in the second phase. Therefore, our results would support the contention by Hardt et al. [10] that the novel cue was integrated into a cognitive map. However, there are a number of key differences between our setup and that used by Hamilton and Sutherland, which may explain the difference in findings. First, we provided participants with 8 trials in phase 1 and a further 8 in phase 2. In contrast, Hamiliton and Sutherland had 20 trials in phase 1 (5 blocks with 4 trials/block) and 12 trials in phase 2 (3 blocks of 4 trials). Similarly, Buckley et al. [28] found a blocking effect using 16 stage 1 trials and 12 stage 2 trials. Although it could be argued that participants in the current experiment did not receive enough training to allow interference to occur, we would contend that all participants clearly learned the task in both phases and that the learning curves for our experiment match very closely with that of Hamilton and Sutherland. A second key difference is the number and location of cues used in the two experiments. Our experiment only had 1 cue in phase 1 and added a second cue in phase 2, whereas Hamilton and Sutherland had 4 cues in phase 1 and added an additional 4 in phase 2. With only a single cue, participants were required to use it and pay attention to it. With 4 cues participants may only have used and learned a subset of cues, an argument put forward by Hardt et al. [10]. The cue feature, particularly the location of cues, seems to play a critical role in determining whether blocking occurs or not. For example, Hardt et al. [10] found that removal of cues close to the target impaired performance on the probe trial, whereas removal of cues further away had less of an impact, suggesting an overshadowing effect (again see Deery and Commins, [6]) and that a full cognitive map was not formed as a result.

A strength of the current experiment is that we kept the cues equidistant from the goal location, so distance was not an issue. Furthermore, the cues were on the same side as the hidden target, ensuring that they would be used. Although the same number of cues was used by Deery and Commins [6], one of the cues was located closer to the goal compared to the other. Under such circumstances, we found an overshadowing effect (similar to Hardt et al. [10]), whereby the cue closest to the platform overshadowed the cue further away. More than this, participants seemed to ignore the more distant cue completely when in a compound, and their performance was a chance level during the test phase despite the cues being on the same side of the arena as the goal. Also, when participants had to learn with a single cue further away, they were slower to learn the task compared to nearer cues (see also Commins et al. [33]). In the current experiment, where distance was ruled out as a factor, participants were able to learn the task equally as well with a single cue or with a compound cue, irrespective of other features (shape, size, color). Therefore, we contend that the removal of distance as a cue feature allows cues to be integrated into a cognitive map, but if distance is a feature of the landmark, then rules of associative learning come into play. Therefore, participants can use both associative learning rules and cognitive mapping, but landmark proximity is the determinant of which strategy is used.

However, further work needs to be carried out to help establish this idea. For example, we only used two cues. While keeping distance equivalent, can multiple cues (e.g., 3, 4 or more cues of different sizes, shapes, colors, brightness) be all integrated into a cognitive map as readily, or is there a limit on number? Furthermore, is there a hierarchy of salience characteristics (apart from proximity), or can all cues be integrated readily, irrespective of their characteristics? Given that we used cues of different shapes, sizes and brightness, our results would suggest there is no hierarchy; however, it could be argued that the green triangle and green square were not distinctive enough in our setup. Furthermore, our cues were on the same side as the target; if the cues were at a further distance (e.g., at the opposite side of the arena to the goal—while keeping the cues equidistant to the goal) could these also be integrated into a cognitive map as readily as the nearer cues in the current experiment. Many authors highlight the importance of the shape of the environment [20,21,22,23,24]; how the shape integrates with the various individual cues and into the overall cognitive map is still an open question.

Another finding that emerges from our results suggests that there may be a hierarchy of learning. We found limited extra exploration when a new cue was added during phase 2—Blocking group. This cue was easily integrated into the cognitive map (as suggested above). There was no significant increase in the time required to find the target on the first trial of phase 2 compared to the last trial of phase 1. When two novel cues replaced a single cue, there was additional learning (Control Group 2), as represented by a small significant increase on the first trial of phase 2. This increase was not as much as when there was no prior learning at all (Control group 1). Therefore, familiarity with the task has a benefit. This general pattern was also observed in the Hamilton and Sutherland [32] study.

Finally, in recent years, there has been strong debate over when blocking might occur. For example, cue type [28], cue stability [11], level of instruction [10], and strategy used [36] all have an impact on whether blocking occurs in spatial learning or not. Furthermore, there has also been much discussion on the nature of the blocking effect more generally (see [37,38]). For example, recently, Maes et al. [37] failed to replicate the blocking effect in 15 experiments. However, there has been much counter-argument to this. Soto [39] argued that the reason for observed failures to replicate was that many of the experiments used the same modality stimuli and were based on outdated models of association [40,41,42]—three contemporary models of associative learning predict a weak or no blocking effect when stimuli are similar or belonging to the same modality [43,44,45]. This is also a consideration in our experiment where the same stimuli of the modality were employed (visual cues) and may add to a reason for our failure to find a blocking effect. The blocking phenomenon is worthy of continued examination and particular focus should be given to what conditions are required for the observation to occur. As this study focused on younger adults, it would be interesting for future studies to investigate blocking in older adults as evidence for cue competition effects is mixed depending on age, with older adults showing strong blocking effects but not overshadowing [46]. While older adults show a similar ability to attend to relevant cues as younger adults [47], it would be worth investigating if this ability declines at a particular point. Additionally, both landmark-based navigation performance decline and preferences for different cue types have been associated with both younger and older age groups, which is worthy of further investigation [48,49].

## 5. Conclusions

Proximity is a key feature in location memory, and there is strong evidence that this follows the rules of associative learning; near cues compete with other cues, near cues overshadow cues further away [6] and they block the learning of any additional cues [50]. However, our findings suggest that if proximity is ruled out as a cue feature, then cues, irrespective of size, shape, or color are treated equally and are integrated into a cognitive map. Therefore, spatial learning follows both associative learning and cognitive mapping rules, but the approach used depends on cue proximity.

## Figures and Tables

**Figure 1 brainsci-15-00414-f001:**
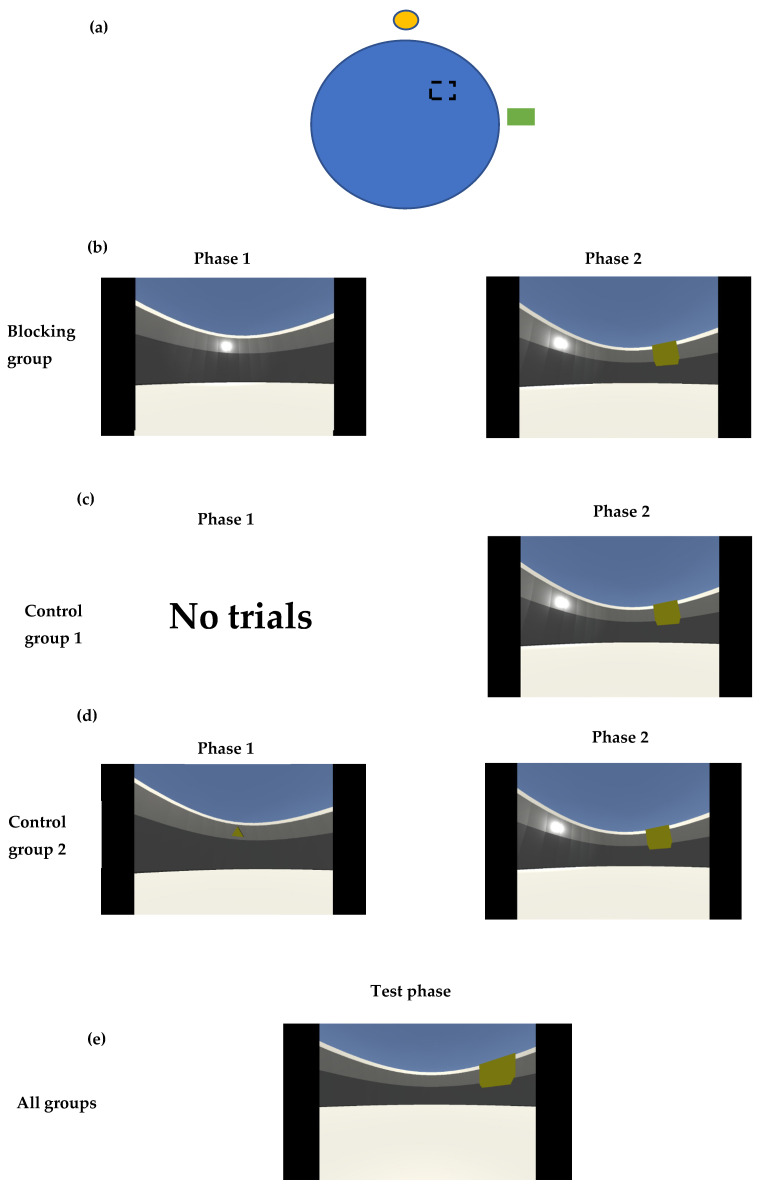
(**a**) Schematic diagram of the NavWell arena. (**b**) Learning phase: Participants in the Blocking group learned using bright light only in phase 1 and bright light and green square in phase 2. (**c**) Participants in the Control group 1 had no trials in phase 1 and learned using bright light and a green square in phase 2. (**d**) Participants in Control group 2 learned using a small green triangle in phase 1 and a bright light and green square in phase 2. (**e**) Test phase. All groups used a single cue (a green square on the eastern wall) to locate the target.

**Figure 2 brainsci-15-00414-f002:**
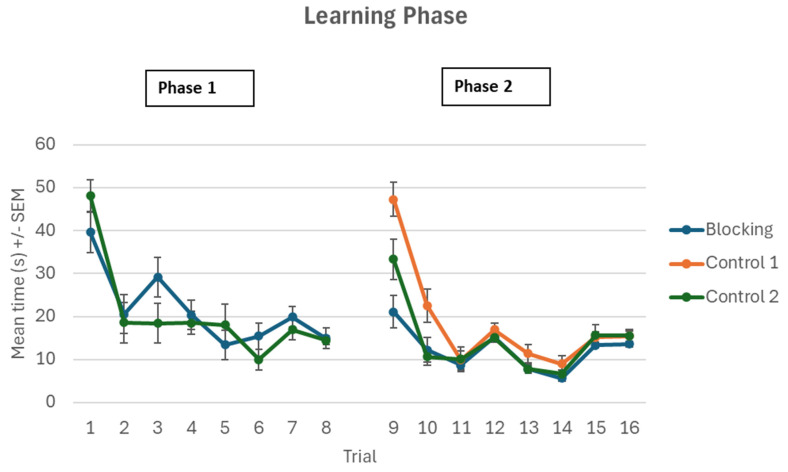
Mean time (+/−SEM) taken by participants find the hidden target in Phase 1.

**Figure 3 brainsci-15-00414-f003:**
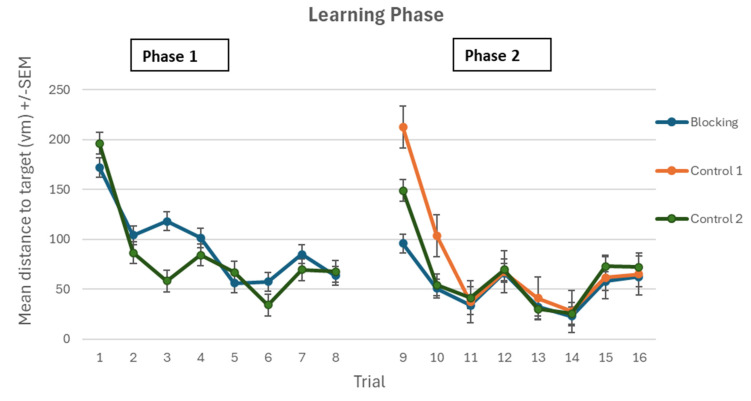
Mean path length (+/−SEM) taken by participants to find the hidden target in Phase 1 and Phase 2.

**Figure 4 brainsci-15-00414-f004:**
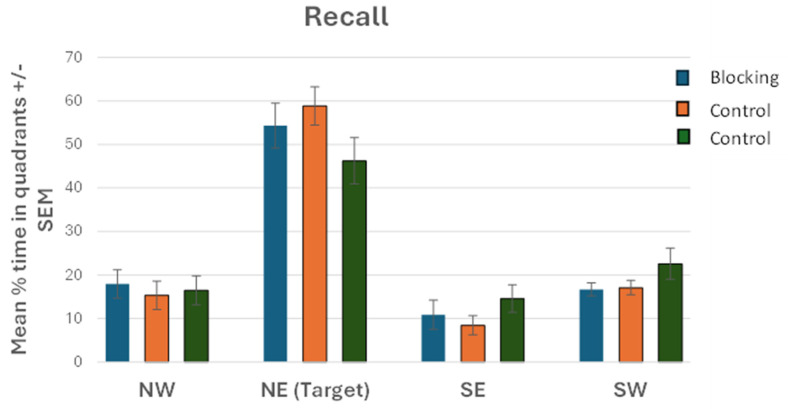
Mean % of time (of 60 s) (+/−SEM) spent by groups in each quadrant during the test phase.

**Table 1 brainsci-15-00414-t001:** Demographic Information for each Experimental Group in the Blocking Experiment.

Group	N	Age (+/−SEM)	M/F	TMT b-a (+/−SEM)
Blocking	20	22.9 (1.1)	10/10	19.6 (1.5)
Control 1	20	20.3 (0.4)	5/15	18.2 (2.0)
Control 2	20	20.6 (0.3)	8/12	20.7 (1.9)

## Data Availability

The data presented in this study are available on request from the corresponding author.

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
