# Peer review of "Equidistant Landmarks Fail to Produce the Blocking Effect in Spatial Learning Using a Virtual Water Maze Task with Healthy Adults: A Role for Cognitive Mapping?"

_brainsci, 2025, doi:10.3390/brainsci15040414_

Round 1
Reviewer 1 Report
Comments and Suggestions for Authors
Although the experiment suggests a finding regarding the associative qualities of the cues, it will be interesting to discuss whether cognitively the participants perform processes of encoding the exploration space, this would explain the consistency of results regarding the location of the cues, the patterns could include spatial location and distance, which in the experiment seems to indicate that there are no increases associated with the development of these patterns. It will be interesting to see this hypothesis as an extension of the conclusions of the study.
Author Response
We thank the review for highlighting this important point and agree that the whole environment made be encoded cognitively. We have now added this idea into our discussion as suggested.
"Many authors highlight the importance of the shape of the environment [18-22]; how the shape integrates with the various individual cues and into the overall cognitive map is still an open question."
Reviewer 2 Report
Comments and Suggestions for Authors
In the considered manuscript the authors describe the experiment in spatial learning and analyze the obtained data, apparently to compare the cognitive map and the associative learning theories. I am not an expert in these, so it is hard for me to evaluate the originality and significance of the study. However, in any case I cannot recommend accepting the manuscript, since it has rather low technical quality, which makes it very difficult to actually understand the authors' intentions and the way they support their arguments. Moreover, the References section is unsatisfactory and in need of radical updating.
First of all, the reference [5] is by the same authors, in the same journal, and has a rather similar title. The authors must clearly explain the difference of the current manuscript with their previous publication.
The Introduction section should be re-written. The part that describes the methodology used by the authors should go to the Methods. The related work review should be better structured, whereas the end of the section must state the research problem and the authors' contribution(s).
The Methods section is also unconventional and incomplete. First, there is no point in having 2.1 level, as it has only one sub-section in it. Second, merging "Participants and materials" appears weird, particularly since there's no description of the materials, but rather of the task (so it would make sense to move it to 2.1.2.). Finally, the authors must add the Design sub-section, detailing the design of the experiment, the variables, the hypotheses, etc.
The Results similarly have problems with structuring. The Demographics (Table 1) seem to rather belong to the Participants sub-section in Methods. In any case, 3.1.1 and 3.1.2, etc. do not belong to Demographics. The presentation of the actual results also needs improvement, as currently a lot of statistical tests are mixed together and it is very hard to grasp (in the absence of explicit hypotheses) what the authors analyze and why.
In the Discussion, one paragraph has over 40 lines of text. Again, better structuring is needed.
Author Response
First of all, the reference [5] is by the same authors, in the same journal, and has a rather similar title. The authors must clearly explain the difference of the current manuscript with their previous publication.
The reviewer is correct, the current paper is a direct follow-up to our previous one. There are two important features of associative learning, overshadowing and blocking. In the previous paper we examined overshadowing in human spatial learning and found an overshadowing effect (proximal cues overshadow distal cues). This suggests an associative account of learning. In the current paper we wanted to examine the second feature – blocking. Interestingly, as reported in the current paper we did not see a blocking effect. This would suggest a cognitive mapping account of spatial learning rather than an associative account. To be honest we are not expecting this, and we are unsure of where this leaves us theoretically. But we have tried to explain the difference in our discussion (now highlighted - Lines 398-412).
The Introduction section should be re-written. The part that describes the methodology used by the authors should go to the Methods. The related work review should be better structured, whereas the end of the section must state the research problem and the authors' contribution(s).
We thank the reviewer for these valid points. We have now re-vamped our Introduction along the lines suggested. (1) We have removed the details of the experimental design and have put this into the Methods. (2) We have also re-structured our introduction by removing many of the studies that focused on blocking in the animal literature. These are probably not relevant to the current experiment. We have also removed some detailed description of a few experiments - again there are not really needed. (3) In addition, as suggested, we have highlighted the problem that we wanted to address in this experiment and made our hypothesis more explicit.
“Given the relative lack of research examining blocking in human spatial learning, particularly with respect to individual landmarks and the inconclusive nature of the results to date, we set out to further explore the blocking phenomenon using a virtual water maze task (VWM) [29]. We have previous found an overshadowing effect in spatial learning [5]; if we also showed a blocking effect, such a finding would further lend support to the associative learning theory of spatial learning rather than the cognitive map theory. Using a paradigm similar to Hamilton & Sutherland [28], we hypothesised that we should observe a blocking effect. Specifically, participants are required to learn the location of a target using a single cue in phase 1, then a second cue is added in phase 2, we hypothesise that when this group is subsequently tested with only the additional cue, then the performance of this group (blocking group) will be impaired compared to two control groups”
The Methods section is also unconventional and incomplete. First, there is no point in having 2.1 level, as it has only one sub-section in it. Second, merging "Participants and materials" appears weird, particularly since there's no description of the materials, but rather of the task (so it would make sense to move it to 2.1.2.). Finally, the authors must add the Design sub-section, detailing the design of the experiment, the variables, etc.
We thank the reviewer for highlighting the formatting issues in our Methods section. As suggested, we have now (1) changed the different levels. We have simply used 2.1, 2.2, etc. (2) We have also removed ‘Materials’ as the reviewer correctly points out we have not described any. (3) We have now also included a Design section (labelled as 2.3) as suggested.
“2.3. Design
This study involved 2 learning phases (learning phase 1 and learning phase 2), where an additional cue was added in learning phase 2. Data from each learning phase were analysed separately. Within each phase a mixed factorial design was used where Trial in (8 trials) was the within group factor and Group (2 groups in learning phase 1: Blocking and Control Group 2; 3 groups in learning phase 2: Blocking, Control group 1 and Control group 2) was the between group factor. Time and distance to the target in each trial were both used as dependent variables for each of the learning phases. Hence, the design was a 2x8 mixed factorial in learning phase 1 and 3x8 mixed factorial in learning phase 2.
In the test phase, Quadrant (4 levels; NE, SE, SW, NW) was the within group factor and Group (3; Blocking, Control group 1 and Control group 2) was again used as the between group factor. Mean percentage time (of 60 seconds) spent in each quadrant was used as the dependent variable. Hence, for the test phase the design was a 4x3 mixed factorial.”
The Results similarly have problems with structuring. The Demographics (Table 1) seem to rather belong to the Participants sub-section in Methods. In any case, 3.1.1 and 3.1.2, etc. do not belong to Demographics. The presentation of the actual results also needs improvement, as currently a lot of statistical tests are mixed together and it is very hard to grasp (in the absence of explicit hypotheses) what the authors analyze and why.
Again, we thank the reviewer for highlighting the formatting issues in our Methods section. As suggested, we have now removed 3.1.1 etc and changed the different levels and their titles (see 3.1, 3.2 etc.). We also used two dependent measures during the learning phase (latency to reach the target and distance to target). Although both measures are related and show a similar pattern of results, we believe it’s important to show that all groups have learned the task using a variety of means. We accept that this might be overkill, and we are happy to remove one of those measures (e.g. distance) should the reviewer deem it important.
In the Discussion, one paragraph has over 40 lines of text. Again, better structuring is needed.
We thank the reviewer for highlighting this. We have now re-structured the Discussion better. In addition, we have also highlighted the discussion between associate learning and cognitive mapping theories better and how the current results on blocking and our previous work on overshadowing fits into the debate. We have also added a conclusions section.
Proximity is a key feature in location memory and there is strong evidence that this follows the rules of associative learning; near cues compete with other cues, near cues overshadow cues further away [5] and they block the learning of any additional cues [42]. However, our findings suggest that if proximity is ruled out as a cue feature, then cues irrespective of size, shape or colour are treated equally and are integrated into a cognitive map. Therefore, spatial learning follows both associative learning and cognitive mapping rules but the approach used depends on cue proximity.
Round 2
Reviewer 2 Report
Comments and Suggestions for Authors
I have read the authors' replies to my previous comments and the revised version of the manuscript. I am glad to see that the authors have addressed the formatting issues in a mostly satisfactory manner.
The problem I noted with the references in the manuscript seems to persist though. I am copying my related comment from the previous review:
"References are deeply outdated, even considering that the field of the study is not a very dynamic one. Not counting the publications by the same authors, out of 40 there are 10% from the last 5 years, less than 25% from the last 10 years!"
After the references are updated (of course, the review of the SotA and the discussion needs to be updated too), I don't mind accepting the paper.
Another minor thing: the Table 1 appears broken in the PDF.
Author Response
We thank the reviewer for highlighting this. The reviewer is correct in the assessment that Blocking in the spatial domain is a relatively under explored area and there has been limited research, particularly in recent times. Despite this, we have gone back into the literature and have added as many relevant recent references as appropriate. Not counting papers from ourselves (out of 48 citations in total) 17 are since 2015 (35%) and 11 are since 2020 (23%). We again thank the reviewer for all his/her comments and believe that our paper has improved substantially as a result of the feedback.